# SPIKE-INSPIRED RANK CODING FOR FAST AND ACCURATE RECURRENT NEURAL NETWORKS

**Alan Jeffares**[1]
University College London, UK
alan.jeffares.20@ucl.ac.uk

**Qinghai Guo**
ACS Lab, Huawei Technologies,
Shenzhen, China
guoqinghai@huawei.com

**Pontus Stenetorp**
University College London, UK
pontus@stenetorp.se

**Timoleon Moraitis**[*]
Huawei Technologies – Zurich, Switzerland
timoleon.moraitis@huawei.com

## ABSTRACT

Biological spiking neural networks (SNNs) can temporally encode information in their outputs, e.g. in the rank order in which neurons fire, whereas artificial neural networks (ANNs) conventionally do not. As a result, models of SNNs for neuromorphic computing are regarded as potentially more rapid and efficient than ANNs when dealing with temporal input. On the other hand, ANNs are simpler to train, and usually achieve superior performance. Here we show that temporal coding such as rank coding (RC) inspired by SNNs can also be applied to conventional ANNs such as LSTMs, and leads to computational savings and speedups. In our RC for ANNs, we apply backpropagation through time using the standard real-valued activations, but only from a strategically early time step of each sequential input example, decided by a threshold-crossing event. Learning then incorporates naturally also *when* to produce an output, without other changes to the model or the algorithm. Both the forward and the backward training pass can be significantly shortened by skipping the remaining input sequence after that first event. RC-training also significantly reduces time-to-insight during inference, with a minimal decrease in accuracy. The desired speed-accuracy trade-off is tunable by varying the threshold or a regularization parameter that rewards output entropy. We demonstrate these in two toy problems of sequence classification, and in a temporally-encoded MNIST dataset where our RC model achieves 99.19% accuracy after the first input time-step, outperforming the state of the art in temporal coding with SNNs, as well as in spoken-word classification of Google Speech Commands, outperforming non-RC-trained early inference with LSTMs.

## 1 INTRODUCTION

Neuromorphic computing is the study and use of computational mechanisms of biological neural networks in mathematical models, software simulations, and hardware emulations, both as a tool for neuroscience, and as a possible path towards improved machine intelligence (Indiveri, 2021). In fact, much of the recent progress in machine learning (ML) is attributed to artificial neural networks (ANNs), which share certain characteristics with biological neural networks. These biological analogies of ANN models include a connectionist graph-like structure (Rosenblatt, 1958), parallel computing over multiple synaptic weights and neurons (Indiveri & Horiuchi, 2011), and the non-von Neumann collocation of memory and processing at each synapse and neuron (Sebastian et al., 2020). On the other hand, state-of-the-art (SOTA) ANNs for ML often miss several other neuromorphic mechanisms that are fundamental in biological neural systems. A characteristic example is that of "spikes", i.e. the short stereotypical pulses that biological neurons emit to communicate (Maass, 1997; Ponulak & Kasinski, 2011). This is a principal neuromorphic feature, characterizing the

---

[1]Work done during project at Huawei – Zurich.
[*]Corresponding author

brain and a category of bio-plausible models known as spiking neural networks (SNNs), but not the most successful ANNs, suggesting unexploited potential and limited understanding of spike-based approaches. Despite the stereotypical, unmodulated shape of spikes, spiking neurons can encode continuous values, for example in their average firing rate (Brette, 2015), which is abstracted into the continuous activation of conventional artificial neurons (Pfeiffer & Pfeil, 2018). Perhaps more interestingly, spikes can also carry information in their specific timing, i.e. through temporal coding that modulates *when* spikes are fired (Brette, 2015). Partly because individual spikes can encode information sparsely and rapidly, biological nervous systems and spiking neuromorphic systems can be extremely energy-efficient and fast in the processing of their input stimuli (Qiao et al., 2015; Davies et al., 2018; Zhou et al., 2021; Yin et al., 2021). This efficiency and speed are key motivations for much of the research on SNNs for potential applications in ML and inference. Moreover, owing to neuronal and synaptic dynamics, SNNs are more powerful computational models than certain ANNs in theory (Maass, 1997; Moraitis et al., 2020). In practice, SNNs have recently surpassed conventional ANNs in accuracy in particular ML tasks, by virtue of short-term synaptic plasticity (Leng et al., 2018; Moraitis et al., 2020). Spike coding itself can also add computational power to SNNs by increasing the dimensionality of neuronal responses (Izhikevich, 2006; Moraitis et al., 2018). However, in practical ML terms, firstly, spike coding poses difficulties to precise modelling and training that require ad hoc mitigation (Mostafa, 2017; Pauli et al., 2018; Pfeiffer & Pfeil, 2018; Woźniak et al., 2020; Comşa et al., 2021; Zhang et al., 2021). Secondly, SNNs are particularly difficult to analyse mathematically and rigorous ML-theoretic spiking models are scarce (Nessler et al., 2013; Moraitis et al., 2020). Thirdly, they require unconventional neuronal models, which do not fully benefit from the mature theoretical and practical toolbox of conventional ANNs (Bellec et al., 2018; Woźniak et al., 2020; Comşa et al., 2021). As a result, the efficiency and speed benefits of temporal coding for ML have been hard to demonstrate with SOTA accuracy in real-world tasks. For instance, very recent literature on temporal coding with SNNs (Comşa et al., 2021; Zhang et al., 2021; Zhou et al., 2021; Mirsadeghi et al., 2021; Göltz et al., 2021) uses rank coding (RC), i.e. the temporal scheme where the first output neuron to spike encodes the network's inferred label (Thorpe & Gautrais, 1998), and it applies it to speed up inference on tasks such as hand-written digit (MNIST) (Lecun et al., 1998) recognition. However, when applied (Zhou et al., 2021) to more difficult datasets such as Imagenet (Deng et al., 2009), the accuracy is significantly lower than in non-spiking versions of the same network (Szegedy et al., 2015). Moreover, in these demonstrations there are no directly measured benefits compared to non-spiking ANNs.

Even though temporal coding is usually not described in terms comparable with non-spiking ANNs, there are in fact ANN architectures with certain analogies to RC, when viewed from a particular angle. Namely, in an SNN that receives a sequence example of several input steps, e.g. in a sequence-classification task, one implication of RC is that the computation for each sequence example can be halted after the first output neuron has spiked and produced an inferred label, even if several steps of the input sequence still remain unseen. Therefore, this is an adaptive type of processing that dynamically chooses the time and computation to be dedicated to each sequence. From this perspective, RC is related to ANN techniques such as Self-Delimiting Neural Networks (Schmidhuber, 2012), Adaptive Computation Time (Graves, 2016), and PonderNet (Banino et al., 2021), which are also concerned with *when* computation should halt. However, these methods do not aim to adaptively *reduce* the processed length of an input *sequence* as RC does, but rather to adaptively *add* time-steps of processing to *each step* of the input sequence. A possibly more deeply related method is that of adaptive early-exit inference (Laskaridis et al., 2021). In this case, the forward propagation of an input throughout layers in a deep neural network at inference time is forwarded from an early layer directly to a classifier head, skipping the layers that remain higher in the hierarchy, if that early layer crosses a confidence threshold. In certain cases, early exit has been applied to sequence classification with recurrent neural networks (RNNs), where the threshold-crossing dictates when in the sequence of an input example the network should output its inference, saving indeed in terms of time and computation (Dennis et al., 2018; Tan et al., 2021). This timing decision based on the first cross of a threshold is similar to inference with RC. However, these early-exit models were not specifically trained to learn a rank code. It is conceivable that this mismatch between training and inference is suboptimal. In addition, this non-RC training of early-exit inference models does not apply the computational savings and speed benefits also to the training process.

Taking together the limitations and benefits of SNNs and of other approaches, what is needed is a strategy that introduces aspects of RC into conventional, non-spiking ANNs, including during training. This would potentially reap the speed and efficiency benefits of this neuromorphic scheme,

without abandoning the real-world usability and performance of ANNs. In addition, these insights into neural coding could feed back to neuroscience. Here we describe and demonstrate such a strategy for temporal coding in deep learning and inference with ANNs. The general concept is simple. Namely, even though ANN activations are continuous-valued and therefore neurons do not *have* to rely on time to encode information as in SNNs, ANNs too *could* time their outputs and, importantly, they could learn to do so. To achieve this in an RNN such as long short-term memory (LSTM) (Hochreiter & Schmidhuber, 1997), we back-propagate through time (BPTT) from a strategic and early time step in each training example's sequence. The time step is decided by the rank-one, i.e. first, output neuron activation to cross a threshold. As a result of this Rank Coding (RC) during training, the network learns not only to minimize the loss, but implicitly also to optimize its outputs' timing, reducing time to insight as well as computational demands of training and inference. In our experiments, we provide several demonstrations, with advantages compared to SNNs from the literature as well as compared to conventional ANNs, including in MNIST classification and speech recognition. Moreover, we show that our method could be applied to SNNs directly as well.

## 2 DEEP LEARNING OF RANK CODING

---
**Algorithm 1** RC-training
---
Given: a training set of $N$ example sequences $\boldsymbol{S}_i = \{\boldsymbol{x}_{i0}, ..., \boldsymbol{x}_{iT}\}$ and corresponding labels $\boldsymbol{y}_i$; an RNN $\mathcal{R}$; and a threshold $\theta$.

1:   $i = 0$
2: **while** $i < N$ **do**                                 ▷ iterate over training examples
3:       $i++$;     $t = 0$;     $T_i =$ duration of sequence $\boldsymbol{S}_i$
4:       $t_{sp,i} \leftarrow T_i$                             ▷ latest possible first "spike"
5:       **while** $t < t_{sp,i}$ **do**            ▷ iterate through input sequence steps until first spike
6:          $t++$
7:          activation $\hat{\boldsymbol{y}}_{it} = \mathcal{R}_t(\boldsymbol{S}_i)$
8:          **for all** output neurons $\hat{y}_j$ **do**
9:             **if** $\hat{y}_{jit} \geq \theta$ **then**                   ▷ Inferred label=$j$
10:             $t_{sp,i} \leftarrow t$                 ▷ rank-one spike time
11:             $\hat{\boldsymbol{y}}_i = \hat{\boldsymbol{y}}_{it}$      ▷ activation at $t_{sp}$ is considered as $\mathcal{R}$'s overall output from $\boldsymbol{S}_i$
12:             BPTT($\mathcal{R}$, Loss($\hat{\boldsymbol{y}}_{it}, \boldsymbol{y}_i$))          ▷ BPTT from $t_{sp}$ only
13:             break                 ▷ Done with this sequence
---

Algorithm 1 shows the RC-training process. The network decides the rank-one spike timing $t_{sp}$ of its outputs based on its output-layer activations and a threshold $\theta$ (line 10). The loss function can be a common on such as cross-entropy. There is no explicit parametrization on time, and only one instantaneous output $\hat{\boldsymbol{y}}_{it}$ is used in the loss (line 12). Therefore, RC-training does not explicitly optimize the timing $t_{sp}$ of the network's outputs. Thus, it is not obvious that learning of the timing aspect can emerge. Nevertheless, timing is implicitly, albeit indeed optimized, as seen in what follows. With random initialization, the activations are nearly uniformly distributed across output neurons, and are smaller than the threshold, throughout the sequence. Without an earlier cross of a threshold, the algorithm applies BPTT from the last step of the sequence (line 4: $t_{sp} = T$). As training advances, minimizing the error between the outputs and the labels minimizes the entropy of the output distribution, i.e. causes the activations at the end of each sequence example to be concentrated around one of the output neurons, such that the maximum activation $\hat{y}_{t_{sp}}^{max}$ is maximized. Through BPTT from that last time step, credit is assigned to earlier time steps as well. Progressively through training, this causes outputs to cross the threshold earlier and earlier, under the condition that relevant, credit-assigned input signal does exist earlier. This conditional acceleration causes the optimization of output timing. This is also shown mathematically in Appendix A.

Importantly, the insight that it is through the minimization of entropy (Eq. 4) that timing is minimized, gives us access to a mechanism for balancing between minimizing the loss and minimizing the timing. Specifically, we introduce to the loss function a regularization term, weighted by a hyperparameter $\beta$, such that minimization of the loss rewards entropy $H$ of the outputs $\hat{\boldsymbol{y}}_{t_{sp}}$:

$$\mathcal{L}_{RC}(\hat{\boldsymbol{y}}_{t_{sp}}, \boldsymbol{y}) = \mathcal{L}(\hat{\boldsymbol{y}}_{t_{sp}}, \boldsymbol{y}) - \beta H(\hat{\boldsymbol{y}}_{t_{sp}}). \tag{1}$$

RC-inference after RC-training is similar to Algorithm 1 but the backward pass (line 12) is not applied. It should be noted that the inference stage on its own performed in this manner, where a threshold decides the timing of the output, is a version of what has been called early exit or early inference. In our implementation, the expectation is that the model learns to encode information in the rank order of its output's timing, because that timing is integrated into the learning process. In this sense, at inference, the model does not merely exit early, but it does so through an underlying learned rank code (RC). Our experimental demonstrations confirmed that the loss is minimized through RC-training, although BPTT's application time-step varies between training examples, and that timing is also minimized down to an optimal floor, as the conditional arguments above (and Eq. 8 in the Appendix) predict.

## 3 EXPERIMENTAL DEMONSTRATIONS

### 3.1 CONTINUOUS SEQUENCE SPOTTING

The first task where we tested RC involves Poisson spike trains of a constant average firing rate of 0.5 spikes per time-step. Each Poisson input sequence consists of 25 binary time-steps (Fig. 1A). The task is to conclude as soon as possible within each sequence whether the sequence includes a period of at least five consecutive time-steps that are all spiking or all silent. That is the positive class (Fig. 1A, blue), whereas the negative class includes at most four consecutive ones or zeros (Fig. 1A, red). We trained an LSTM model on this task with cross-entropy loss and RC. When the positive-

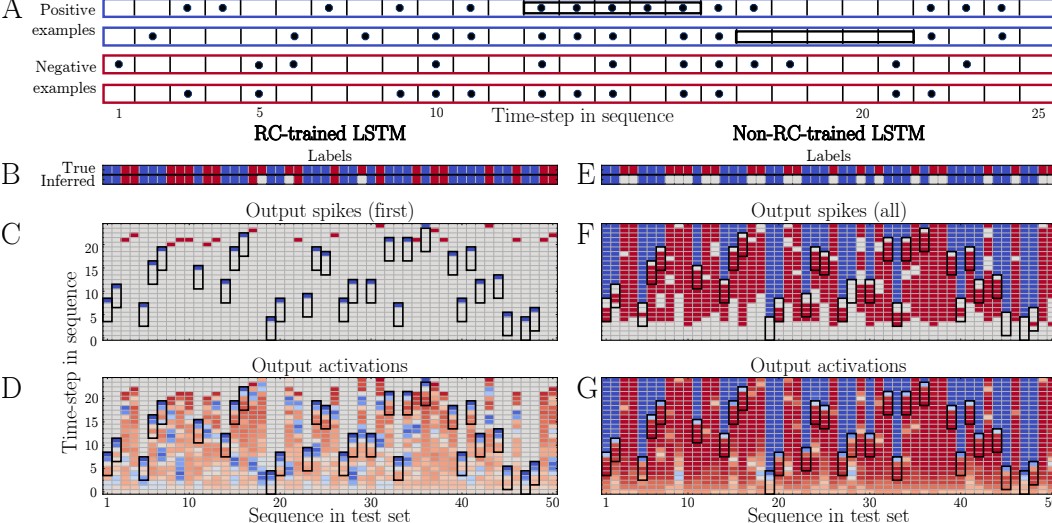

Figure 1: Continuous sequence spotting. A: Example spiking inputs. Black rectangles indicate five spiking or silent continuous time-steps implying a positive example (blue, not red). B-D: RC-trained inference. B: Labels. C: first output spikes (blue, red, and grey indicate positive, negative, or no spike). D: Activations until first spike. E-G: Non-RC-trained model (backpropagation from the end of each sequence). In F and G the model was allowed to continue operating after the first spike.

class-related output activation crossed a threshold, we considered this a positive-class output spike, and vice versa for the negative class. That first spike's label was used as the inferred label from the model for each input sequence (Fig. 1B), as described by RC inference. In few cases, there was no output spike produced (Fig. 1B, grey). These were specifically of the negative class, so they could be considered as negative inferred labels. The RC-trained model achieved 100% accuracy, confirming that training did work in terms of minimizing the explicit loss. In addition, the positive spikes occurred in almost all cases at the earliest possible time-step relative to the five-step subsequences (Fig. 1C, blue spikes & black rectangles), showing that RC-training did optimize the timing aspect of the task as well. The timing effect is visible also in negative examples, where the absence of a five-step-long continuous subsequence was recognized earlier than the last time step. The importance of RC during training becomes clear if we compare to the result of training the same network without

RC. That is, we trained it with BPTT from the last time-step of each training example. In this case, we could perform inference conventionally, at the end of the sequence. However, attempting to perform early, i.e. RC, inference in a manner similar to the RC-trained model is problematic in this case. The first output spike is always negative (red) regardless of the true class, therefore the first spike label cannot be used (Fig. 1F). Label prediction in this case can be improved by a scheme where examples are considered negative unless a positive (blue) spike is output, whereas negative (red) spikes are ignored. In terms of timing, the first positive spikes are in fact emitted soon relative to the observation of a five-step subsequence, but still one step too late (Fig. 1F, bottom-most blue spikes). Moreover, through this scheme, inference of negative labels cannot be performed before the full 25 steps of an input sequence are all observed. Therefore, RC training is necessary for fastest inference, for less computation, and for sparseness of output spikes. To gain deeper insight into the RC-trained network, we also study its activations before the application of the threshold and compare with the conventionally-trained model (Fig. 1D & G). Interestingly, the RC model flexibly adapts its belief throughout each input sequence based on available evidence. In contrast, the non-RC model begins early on in the 25 steps to believe inputs as negative examples, and does not show signs of rapid adaptation to the changing input evidence. This by-default negative (red) belief, and the slow switching of opinion cause comparatively low confidence when the 5-step sequences are present (Fig. 1G, light blue). Furthermore, the sparsity that we observed in terms of output spikes of the RC model is observable also at the pre-spike activations. Both positive and negative activations remain low (light colours in Fig. 1F) until enough evidence appears. In summary, RC-training solved both the classification and the timing aspects of this task, while it also introduced sparsity in the network's activity, and it reduced the computational steps in training and inference.

## 3.2 2-SEQUENCE PROBLEM

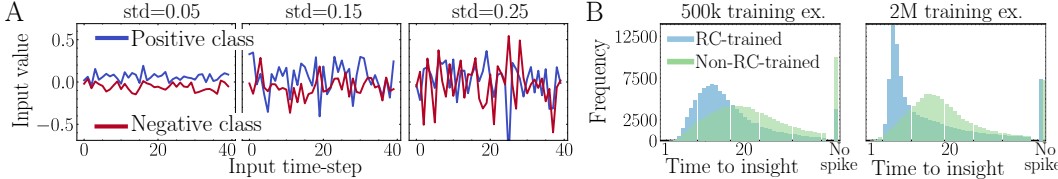

Figure 2: (A) Input examples & (B) inference times of RC & non-RC LSTM throughout training.

The second task is inspired by the original "2-sequence problem" originally introduced in Bengio et al. (1994) and adapted by Hochreiter & Schmidhuber (1997). The task is to classify input sequences into one of two classes. Each example of the positive class is a sequence of 40 independent samples from a Gaussian distribution centred at +0.05, whereas the negative class is centred at -0.05 (Fig. 2A). Each sequence has a constant standard deviation sampled uniformly from the range (0.05, 0.25). These sequential inputs provide progressively more evidence on the underlying mean value, which is difficult or impossible to infer correctly from a short sequence when the noise is high. Here as well, we trained an LSTM model conventionally i.e. by back-propagating from the last time step of each sequence, and compared to our RC-training approach. The RC-trained model, evaluated on a test set at the time of its first spike, i.e. with RC-inference, achieved an accuracy of 95.45%. This is lower than the 96.84% reached by the non-RC-trained model, when evaluated at the last, i.e. 40th, time-step. However, the RC-trained model produced its RC-inference early, after 14 time-steps on average. The accuracy of the non-RC-trained model, if evaluated at the 14th time-step, is much lower (89%). Alternatively, we also tested the non-RC-trained model with RC-inference, and the model achieved a higher accuracy of 96.32%. However, it was significantly slower than the RC-trained model, with an average first spike after 23 time-steps. If we examine the distribution of the first spike time as it changes throughout training, it can be seen that the RC-trained model becomes significantly faster progressively (Fig. 2B, blue), which confirms experimentally our theoretical result (Eq. 8) that RC-training optimizes timing. Conversely, when the non-RC-trained model is evaluated with RC-inference throughout its training, it shows that conventional training has little effect on the timing distribution. In this task, therefore, the RC-trained model achieves faster inference, by optimizing timing, albeit at the expense of accuracy. As we have introduced in the paper's theoretical section, it is theoretically possible to use a regularization that prevents over-optimization of timing at any cost. We demonstrate this in the following more challenging tasks.

### 3.3 SNN BENCHMARKS ON TEMPORAL MNIST

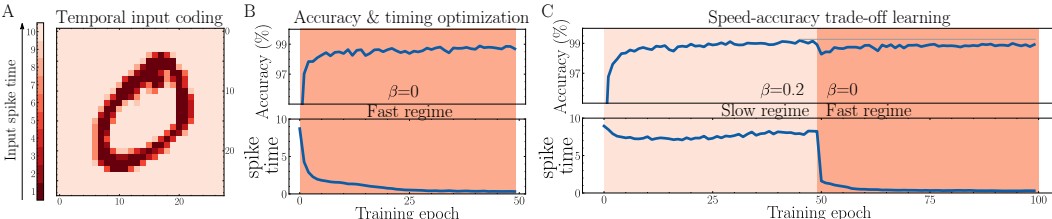

Figure 3: Temporal MNIST with RC-convLSTM. 2 distinct regimes, i.e. an accurate & a fast one.

The MNIST dataset is a common image-recognition benchmark. It includes grey-scale images of handwritten digits, where each pixel is an intensity value between 0 and 255. The task of classifying such digits into 10 classes is considered rather simple by modern ML standards. However, more complex versions have been designed to match and demonstrate the temporal capabilities of SNNs, and contrast them to ANNs, by encoding the input temporally. In a common such version (Comşa et al., 2021; Zhang et al., 2021; Zhou et al., 2021; Mirsadeghi et al., 2021), each input pixel's intensity is encoded by the timing of a single and binary spike in a time interval, where darker pixels spike earlier. Some SNNs operate in continuous time, but here we use discrete time, and we encode each MNIST frame over 10 time steps (Fig. 3A). Some of the recent SNN work has demonstrated that temporally coded SNNs can perform the task earlier than the end of each input sequence (Comşa et al., 2021; Zhang et al., 2021). Of particular background interest is a finding in Comşa et al. (2021). Specifically, temporal coding has two possible regimes when performing this task. One regime is relatively slow but more accurate. The other is a faster but less accurate regime, where on average less than 15% of the input frame's duration has passed before inference. As a result, in this task, for each reported accuracy, the regime of speed must be specified, as is indeed in Comşa et al. (2021). A different, very recent work reports results outperforming those of Comşa et al. (2021), but does not mention the presence of two regimes. It does show histograms of inference time, where it appears that the model is not in the fast regime, and it mentions a training time in the order of 100 epochs, at which point Comşa et al. (2021) described that the network is at a high accuracy-low speed regime. Here, we RC-trained an LSTM and a convolutional LSTM (ConvLSTM) (Xingjian et al., 2015). Firstly, the RC models did learn to perform inference in a very fast regime, on average immediately after the very first input time-step (Fig. 3B). Next, we confirmed that our theoretical method of balancing the speed-accuracy trade-off, through an entropy-rewarding regularization term weighted by $\beta$, is effective. By using a higher $\beta$ during RC-training, the networks remained in a slower but more accurate regime. By switching to a lower $\beta$ value during training, the model switched to the faster but slightly less accurate regime. Therefore, this validates the intended functionality of our regularization term, and also confirms the presence of the two discrete regimes in this task (Fig. 3C). These were observed in every network that we trained, and Fig. 3 B & C shows these effects in one of the models, namely a hidden layer of 20 ConvLSTM units, all-to-all connected to 10 output neurons with a softmax. The speed-up is attributable to RC-training (see Appendix B.3). Our models outperformed the SOTA SNNs in both the fast and the slow regime, except much larger and deeper SNNs that slightly outperformed ours. Our models and their accuracies compared to the literature are presented in Table 1. In addition, we could achieve high accuracy in the fast regime within only 50 epochs (Fig. 3B), while the existing SOTA in the fast regime required several hundreds of training epochs (Comşa et al., 2021)). For all results, we used Adam with a learning rate fixed at 0.001, and the threshold of 0.95. The only hyperparameter value that we searched systematically was $\beta$. These results are noteworthy because they achieve and surpass some of the advantages of SNNs by using a spike-based technique, but avoiding the main difficulties of SNNs. Moreover, RC enables a novel reduction of the resource requirements. Specifically, it reduces the computation per training example or batch, as RC only backpropagates from the first output spike. In contrast, even in the temporal coding SNN literature, all output neurons spike before backpropagation is applied. Notably, our new RC method is applicable to SNNs (Appendix E).

| | Reference | Accuracy | Architecture | Model |
|---|---|---|---|---|
| < 15% of frame (Fast regime) | Comşa et al. (2021) | 97.4 | 784-340-10 | $\alpha$-PSP |
| | Comşa et al. (2021) | <97.4 | 784-1000-10 | $\alpha$-PSP |
| | **RC-training (this work)** | **98.14** | 784-340-10 | RC-LSTM |
| | **RC-training (this work)** | **98.90** | 784-10-10 | RC-ConvLSTM |
| | **RC-training (this work)** | **99.19** | 784-20-10 | RC-ConvLSTM |
| > 25% of frame (Slow regime) | Comşa et al. (2021) | 97.96 | 784-340-10 | $\alpha$-PSP |
| | Zhang et al. (2021) | 98.1 | 784-340-10 | ReL-PSP |
| | Zhang et al. (2021) | 98.5 | 784-800-10 | ReL-PSP |
| | Zhang et al. (2021) | 98.1 | 784-1000-10 | ReL-PSP |
| | **RC-training (this work)** | **99.16** | 784-10-10 | RC-ConvLSTM |
| | **RC-training (this work)** | **99.29** | 784-20-10 | RC-ConvLSTM |
| Slow regime & larger network | **Zhang et al. (2021)** | **99.4** | 784-16Conv-P2- -32Conv-P2-800-128-10 | ConvReL-PSP |

Table 1: Comparison of our work to recent SNN literature on temporally-coded MNIST.

### 3.4 RAPID KEYWORD CLASSIFICATION - GOOGLE SPEECH COMMANDS

The last and most advanced task that we addressed is a form of speech recognition. Specifically, we used the Google Speech Commands dataset v0.02 (Warden, 2018). This is a popular dataset containing 105,829 utterances of 35 different spoken terms from 2,618 speakers, and each of these recorded segments is at most 1-second long. 10 of the spoken terms ("Yes", "No", "Up", "Down", "Left", "Right", "On", "Off", "Stop" and "Go") are selected as keywords, whereas the remaining 25 terms are combined into an 11th, "unknown" class. We applied a preprocessing that is standard for this type of task, generating log-mel filterbank energies to extract constituent frequencies of the recording, and a moving window. This resulted in 20 frequency-features over a sequence of 81 frames. We used a network with 128 LSTM units before two fully connected layers of 32 and 11 neurons each, with a softmax at the output. The training, validation and testing split of the dataset followed the convention provided in section 7 of Warden (2018). We trained the model with RC, examining whether RC is effective in this task too, and then we compared to a baseline early inference after conventional training with BPTT from the end of each training sequence-example. First,

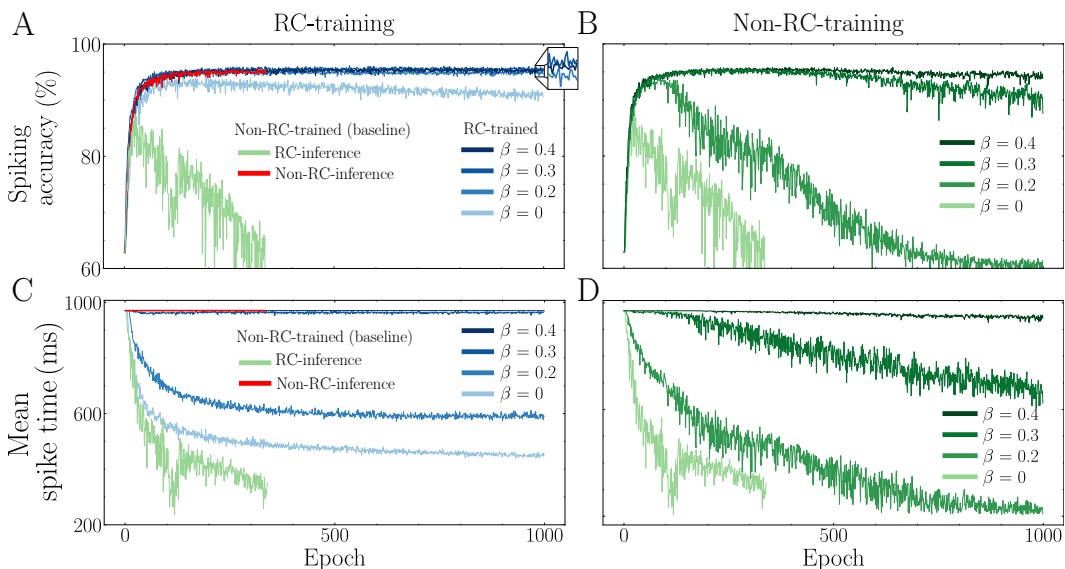

Figure 4: Validation accuracies (top row) and spike times (bottom row) during RC- (left) and conventional (right) training on Google Speech Commands, for varying regularization parameters. The left panels (RC-training) include a non-RC-trained baseline.

throughout training for 1000 epochs we looked into the accuracy of the model on the validation set,

evaluated using RC-inference with the same spiking threshold (0.95) as was used for RC-training (Fig. 4A and C). In parallel, we recorded the mean time-to-inference on the validation set, as defined by the first output spike time. In one case we used standard cross-entropy loss, i.e. with the confidence-regularization term $\beta$ set to zero. We observe that, in this case, spike time is significantly reduced by RC-training, down to less than 500 ms on average, compared to the 1000 ms of the full speech sequences (Fig. 4C lightest blue curve). In fact, training continues to minimize the spike time. However, the validation accuracy (Fig. 4A, lightest blue) also decreases with training. This is another expression of the effect that we observed in previous tasks, where continued minimization of the timing trades off accuracy. To resolve that, we introduce a non-zero $\beta$ regularization term, which achieves its goal to balance timing minimization against loss minimization. Specifically, a $\beta \leq 0.3$ increases the RC-inference validation accuracy compared to $\beta = 0$. However, it keeps the mean spike time delayed until the end of the sequence (Fig. 4A & C, darkest blue). An intermediate value of $\beta = 0.2$ (medium-light blue curve) achieves a much better trade-off, where the validation accuracy is almost identical to the one achieved by $\beta = 0.4$, but validation inference speed is significantly higher, with a stably converged spike time of around 600 ms. As a first baseline, we compare these validation accuracies and speeds to a model trained with backpropagation from the last time-step of each sequence and without our regularization (i.e. $\beta = 0$). This non-RC-trained model, validated using RC-inference, reached a maximum accuracy that was lower than the accuracy of the best RC-trained models, and subsequently deteriorated fast (Fig. 4A, light green). Importantly, even when evaluated, not on RC spiking early inference, but on conventional end-of-sequence inference, its validation accuracy (Fig. 4A, red curve) is not significantly higher than the RC-trained RC-inference – the curve is rather indistinguishable. This suggests that RC-training optimizes accuracy comparatively well, despite also focusing on the timing. We will quantify this comparison to non-RC training and make it conclusive on the test set, but first we will further analyse the non-RC training process. As the best validation performance of the RC-trained models was achieved by incorporating a non-zero confidence-regularization term $\beta$, we explored whether it is this regularization, rather than the RC-training itself, that is responsible for the seeming timing advantage of the RC-trained model. Specifically, we trained with conventional backpropagation from the end of each sequence using cross-entropy but now added regularization to the loss function by $\beta \in \{0.2, 0.3, 0.4\}$, in addition to $\beta = 0$ as mentioned before. We performed threshold-based early inference on the validation set throughout training, using the same confidence threshold (0.95) as for the RC-trained models. Different thresholds were explored in the analysis we describe later. The resulting accuracy and speed curves are shown in Fig. 4. It can be seen that values of increasing $\beta$ improves the convergence of the validation accuracy (Fig. 4B). However, to converge to a stable level of accuracy, a $\beta > 0.3$ is required, which comes at a significant cost of speed. Therefore, $\beta$ regularization alone does not suffice to obtain the benefits of RC-training.

Ultimately, most interesting in such an application is the quality of the resulting trade-off between speed and accuracy on the test dataset. In addition, the tunability of this trade-off is important. It is indeed possible to tune this trade-off by varying the threshold of the trained model, thus obtaining a varying accuracy as a function of inference time, i.e. spike time. This is an existing approach in the literature (Sun et al., 2016; Dennis et al., 2018; Tan et al., 2021). By lowering the threshold, faster inference is obtained for a cost of lower accuracy. As a result, for each trained model, the curve representing the trade-off and its tunability is the curve of accuracy as a function of mean spike time. This curve, for the case of the non-RC trained models, is shown in Fig. 5B. We used the network weights that produced the best validation accuracy throughout training (i.e. the maxima in Fig. 4B). The models that were learned through different $\beta$ values with conventional training had mostly similar trade-off curves among them. To the contrary, in the case of RC-training, the trade-off curves were different for different values of $\beta$. As a result, RC-training offers two methods for choosing the desired trade-off. First, for a given trained model, the spiking threshold can be lowered for faster but less accurate inference (Fig. 5A, each blue curve), as in the conventional non-RC-trained models. However, the accuracy drop by lowering the threshold is significant, in both the RC- and the non-RC-trained cases. RC-training offers a second method, with a smaller accuracy drop for faster inference. By varying $\beta$ while keeping the threshold at a fixed high value, significantly earlier inference is obtained for a much smaller drop or even slight increase in accuracy (Fig. 5A, red curve), as a result of RC-training. Therefore, through RC-training, we have an additional option for tuning the speed of inference, and the model recognizes the spoken keywords with higher accuracy than the non-RC-trained model, at any desired speed setting. It should be noted that the threshold-tuning can be used simultaneously with the $\beta$ tuning method, as the two are orthogonal to each other and combinable.

Hence, more advanced threshold tuning mechanisms can be added after RC-training. For example, it is possible to have a different threshold for each time-step through the input examples, and this set of multiple thresholds at the inference stage can be optimized for a certain increase in accuracy (Tan et al., 2021). Here instead we explored the direction of incorporating timing into the training process. Future work could combine RC-training with the literature's mechanisms that also optimize the RC-inference stage. On $\theta$ during training, see Appendix C. In summary, RC-training was again effective in optimizing the timing of inference, also in the time-critical application of keyword recognition within speech recordings. It resulted in significantly improved speed-accuracy trade-offs, and, in combination with an entropy-based regularizer, it improved the tunability of those trade-offs.

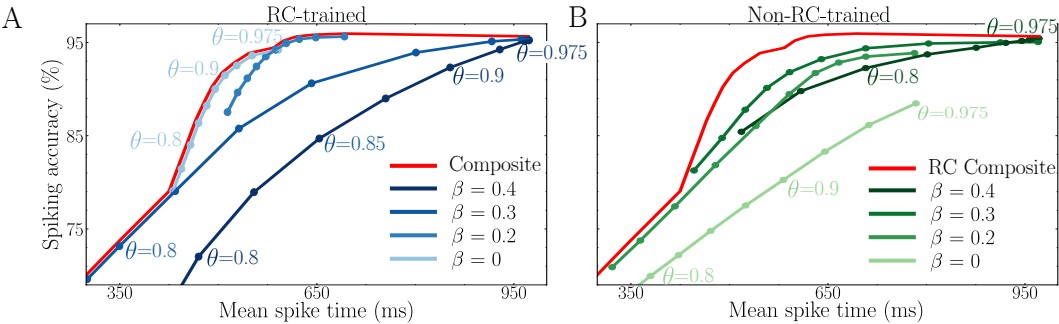

Figure 5: Speed-accuracy trade-off curves of RC- & non-RC-trained models produced by varying the RC threshold $\theta$ at inference, for various values of $\beta$. Red: The resulting composite best trade-off.

## 4 DISCUSSION

All in all, we provided a theoretical motivation and a practical implementation for a spike-inspired technique that adaptively decreases computation and increases speed during both training and inference, while only trading off a small cost of accuracy, and offering flexibility in choosing the desired speed-accuracy trade-off. On the one hand, our results strengthen the case for research in spike-based networks, as they provide direct evidence for the benefits of spikes. This is supported by our comparisons of RC- to non-RC-training. On the other hand, our results suggest that conventional ANNs too can benefit by integrating specific isolated spike-inspired elements. For example, here spikes were used only in the output, and only their triggering aspect was used, while the remaining mechanisms were conventional RNNs and deep learning. This careful isolation of a specific neuromorphic feature allowed us to better understand and demonstrate several of RC's advantages, despite differences from Thorpe & Gautrais (1998). Other recent work also provided direct evidence for the optimality of neuromorphic computing through focused application of very specific mechanisms (Leng et al., 2018; Moraitis et al., 2020). In fact, the approach of identifying atomic neuromorphic computational elements, isolating them from surrounding related complexities and embedding them in otherwise conventional computational models, appears incremental but may be a more pragmatic path towards fully neuromorphic computing. In addition to these broad implications, our results also imply more focused impact, namely on SOTA benchmarks for SNNs, on SOTA algorithms for fast inference, on the growing field of adaptive computation with RNNs, and on speech recognition. Therefore, further exploration of RC-RNNs may benefit several different research directions.

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

# APPENDIX

## A  MATHEMATICAL DERIVATION OF RC-BASED TIME-OPTIMIZATION

In Section 2 we argued that RC (Algorithm 1) optimizes the timing of the network's outputs. Below, after the definitions and prerequisites 2-4, here we show this more concretely.

$$t_{sp} := \{\hat{y}_{t_{sp}}^{max} \geq \theta \quad \wedge \quad (\hat{y}_t^{max} < \theta, \quad \forall t < t_{sp})\} \qquad \text{(t of first threshold-cross).} \tag{2}$$

$$j := \text{class of the input.} \quad \min Loss(\hat{\boldsymbol{y}}_{t_{sp}}, \boldsymbol{y}) \implies P(j|\boldsymbol{x}_0, ..., \boldsymbol{x}_{t_{sp}}) = \hat{y}_{j,t_{sp}} = \hat{y}_{t_{sp}}^{max}. \tag{3}$$

$$\text{Moreover,} \min Loss(\hat{\boldsymbol{y}}_{t_{sp}}, \boldsymbol{y}) \implies \min Entropy(\hat{\boldsymbol{y}}_{t_{sp}}) \implies \max \hat{y}_{t_{sp}}^{max}. \tag{4}$$

$$\Big[ \text{RC-training} \,\Big|\, \text{given that signal that is partly responsible for label } j \text{ is present in the input before } t_{sp} \Big]$$

$$\implies \left[ \min Loss(\hat{\boldsymbol{y}}_{t_{sp}}, \boldsymbol{y}) \quad \Big| \quad \frac{\partial P(j|\boldsymbol{x}_0, ..., \boldsymbol{x}_{t_{sp}})}{\partial P(j|\boldsymbol{x}_0, ..., \boldsymbol{x}_{t_{sp}-1})} > 0 \right] \tag{5}$$

$$\overset{(4)}{\implies} \left[ \max \hat{y}_{t_{sp}}^{max} \quad \Big| \quad \frac{\partial P(j|\boldsymbol{x}_0, ..., \boldsymbol{x}_{t_{sp}})}{\partial P(j|\boldsymbol{x}_0, ..., \boldsymbol{x}_{t_{sp}-1})} > 0 \right] \tag{6}$$

$$\overset{(3)}{\implies} \left[ \max \hat{y}_{t_{sp}}^{max} \quad \Big| \quad \frac{\partial \hat{y}_{t_{sp}}^{max}}{\partial \hat{y}_{t_{sp}-1}^{max}} > 0 \right] \tag{7}$$

$$\overset{BPTT}{\implies} \max \hat{y}_{t_{sp}-1}^{max} \implies \max \left( \hat{y}_{t_{sp}-1}^{max} - \theta \right) \overset{(2)}{\implies} \min t_{sp}. \tag{8}$$

This shows indeed that, through optimizing cross entropy, RC also optimizes the timing down to the minimum timing that satisfies the condition of credit assignment to a previous time step (Eq. 5).

## B  ADDITIONAL DETAILS ON EXPERIMENTS

### B.1  CONTINUOUS SEQUENCE SPOTTING

This experiment allows us to compare RC training to traditional EOS training in a noise-free controlled environment. This is the only experiment in which there is an unambiguous optimal spike time step for each sequence. This allows us to isolate the effect of RC training when evaluation the posterior probabilities and compare against a ground truth in terms of both classification and timing.

Throughout this experiment we use a standard LSTM architecture with a hidden state of size 125 and a projection to a single output neuron followed by a sigmoid activation. We apply binary cross-entropy loss and Adam optimiser Kingma & Ba (2014) with a learning rate of 0.0003. In this experiment we set $\beta = 0$ meaning we do not use a confidence penalty regularisation. Data is generated in batches of size 128 with 1,500,000 training examples in total. An independent validation set of 2000 sequences is evaluated after every 50 training batches. We consider two distinct training approaches: our proposed rank-coded training and EOS training where just the final model output is used for prediction. The task is to produce a model that can robustly identify these continuous sequences as they occur where we consider any output greater than $0.95$ as being a positive spike and any output less than $0.05$ as being a negative spike.

### B.2  2-SEQUENCE PROBLEM

This dataset introduces an imperfect relationship between sequence values and class labels and allows us to consider the exact quantity of Gaussian noise added to a given sequence to be a proxy for its level of difficulty. A desirable property of RC training would be to appropriately trade off speed and accuracy by providing a balance of both fast and accurate predictions. Since each additional element of the sequence provides additional evidence of the sequence class, the optimal choice to maximise accuracy is to predict at the end of the sequence. Therefore this problem allows us to compare RC training to traditional EOS training at trading off speed and accuracy.

In this experiment we use the same LSTM architecture, optimiser and hyperparameters as in the previous section. We consider the same two training methods: RC training and EOS training. We train both models on 2,000,000 training examples in batches of size 128 and retain the model that achieves the highest spiking accuracy on a validation set of size 2000 evaluated every 50 batches.

### B.3 TEMPORAL MNIST

This problem is a standard benchmark for temporal coding schemes in the SNN literature. This allows us to link our work to the existing body of research and compare performance. Previous comparisons to ANNs in these works only considered non-recurrent neural networks which had no temporal aspect to their inference. By introducing this task we can compare our proposed RC training in ANNs to SOTA approaches from the SNN literature while also controlling for speed of inference. This provides a comparison between these fields on a task originally designed to exploit the temporal nature of SNNs.

We compare rank-coded training to SNN benchmarks by training a hidden layer 20 or 10 convolutional LSTM units or 340 LSTM units, and a projection layer of 10 neurons representing the 10 MNIST classes. We also include the $\beta$-weighted confidence penalty regularisation term to our loss calculation for the first time. For a fair comparison to the literature, we follow the experimental protocol taken in Comşa et al. (2021) and Zhang et al. (2021), as follows. Each model is trained on the 60,000 MNIST training examples and test accuracy is reported on the testing set of 10,000 examples. Unlike the large hyperparameter search, e.g. in Comşa et al. (2021), we only required to search over a single hyperparameter $\beta$ over which we completed a random search in the range $[0.15, 2]$ with the LSTM model. We found $\beta = 0.165$ as performing best on LSTM, and we then used it, without a separate search, also for training the convolutional LSTM. With this hyperparameter, we trained each model twice, and we report the top accuracy on the test set.

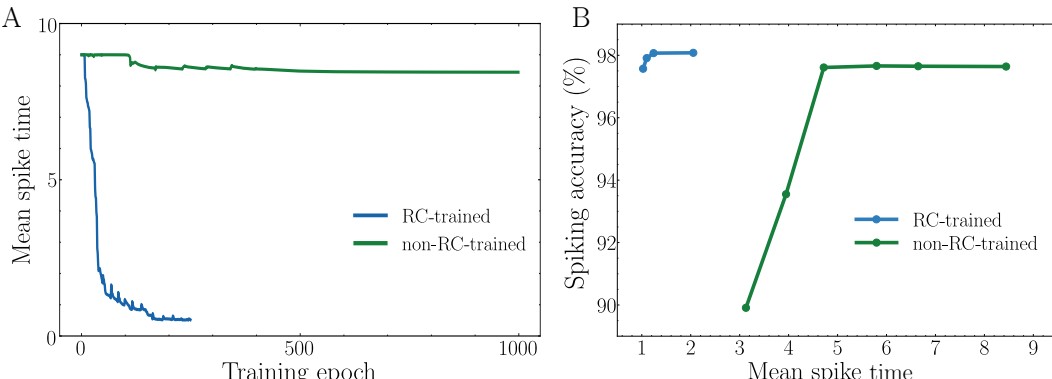

Figure 6: RC vs non-RC training on temporal MNIST. A. The non-RC model does not minimize its timing throughout training, even if tested with RC inference, and even if trained much longer. B. Decreasing the RC-inference threshold speeds up inference, but for very early inference the non-RC model's accuracy drops significantly, whereas the RC model remains accurate despite very early outputs.

**RC is the cause of early inference in Temporal MNIST, not the data or the network alone.** In Section 3.3 we compared an RC-trained LSTM network to the current SOTA benchmarks in the temporal coding literature, showing it achieves fast and accurate inference. A reasonable follow-up question is whether the early inference in this case is not due to the learned RC, but rather only due to particularities in the specific dataset or properties of LSTM that SNN models don't have. To evaluate this, we trained an LSTM network (784-340-10) with RC and compared to non-RC-training. In the non-RC case, we calculated the loss and applied BPTT conventionally, from the end of the sequence, on the 10th and final time step. In Fig. 6A, we present the mean first spike time when both methods are evaluated throughout the training process using RC inference with a fixed threshold of 0.95. The non-RC model does not minimize its timing throughout training, even if tested with RC inference, and even if trained for a much longer time. The final model from each method

was then evaluated on the test set for a range of threshold values as displayed in Fig. 6B. The first-spike accuracy for the RC-trained model is evaluated at threshold values $\theta \in \{0.85, 0.9, 0.95, 0.99\}$, while the non-RC model is evaluated at $\{0.25, 0.35, 0.45, 0.55, 0.75, 0.95\}$. We observe that because the non-RC training objective is to maximize accuracy at the end of each input sequence, it is not incentivised to provide reasonable outputs much earlier in the sequence, even though there is enough exploitable information early. RC-training, on the other hand, is implicitly optimizing the timing of its responses. This provides evidence that the strong performance in this task is largely a function of the training mechanism rather than just a result of the dataset or of using an LSTM architecture.

### B.4 Rapid keyword classification - Google Speech Commands

This task is introduced to evaluate RC training on a popular sequential benchmark from the machine learning literature. Classifying keywords in speech signals is, in principle, an excellent candidate for RC training. In most examples throughout this dataset the correct class can be predicted before the end of the sequence with any frames following the end of the keyword carrying no useful information and thus resulting in wasted computation. This task provides the opportunity to evaluate if RC training can still achieve SOTA accuracy among recurrent models on this task while significantly decreasing inference time.

In this experiment we selected standard choices (Zhang et al., 2017) for all preprocessing parameters using a hamming window function over a window of 25 ms with a stride of 10 ms and extracting 20 mel coefficients without applying padding. We also include an input context of 10 frames on either side of the input frame as is common for RNNs on this dataset, see e.g. Sun et al. (2016). This entire pipeline results in the full one-second sequences being 81 frames in length (after padding of the small fraction of sequences not consuming the full second). Because some sequences in the dataset are shorter, the average sequence length before padding is 78 frames. We also applied $L2$-norm gradient-clipping with a maximum of 0.25. We applied early stopping to any model for which the spiking accuracy fell below the proportion of the majority class (non-keyword class) in the data. This only occurred for the non-RC-trained model with $\beta = 0$.

### C (Non-)effect of the threshold choice $\theta$ during RC training

Throughout this work we considered the threshold $\theta$ during RC training to be a fixed hyperparameter. In the training phase, the trade-off between resulting, i.e. inference, speed and accuracy is controlled by varying $\beta$. We used a value of $\theta = 0.95$ during training. Once the model is trained, thus far, we have shown the possibility of also varying the threshold during inference as an additional mechanism to further tweak this trade-off (Fig. 5).

As an alternative approach, we also considered tuning the threshold during the training phase. We found it to have only a minimal effect on performance. As an example, in Fig. 7, we present the results of varying the threshold during training on the 2-sequence problem (Section 3.2 and Appendix 3). In this experiment, each model is trained as in Section 3.2, with a fixed threshold throughout training. Each of the following candidate threshold values was used: $\{0.85, 0.9, 0.95, 0.99, 0.999\}$. Once trained, the model is tested on 100,000 random sequences using these thresholds, and the resulting trade-off curves for the different RC-training thresholds are shown in Fig. 6. As can be seen in the figure, the difference in performance between different training thresholds is minimal as long as the threshold is reasonably high.

### D Hidden temporal dynamics

Our implementation of temporal coding considers only the first output spike, by applying a simple threshold-based rule. However, the network manages to conform to this rule through more complex operations that implement a more complex temporal code in the recurrently connected population of neurons. Essentially, the simplicity is in the supervisory interface for training the network, but, to map the input sequence to a well-timed and accurate summarizing spike, the network uses a more involved, multidimensional, i.e. multi-neuron, temporal encoding in the neurons' hidden state and cell state. An example of such hidden temporal dynamics can be seen in Fig. 8 in the response of the RC-trained LSTM network to an example recording from the Google Speech Commands dataset.

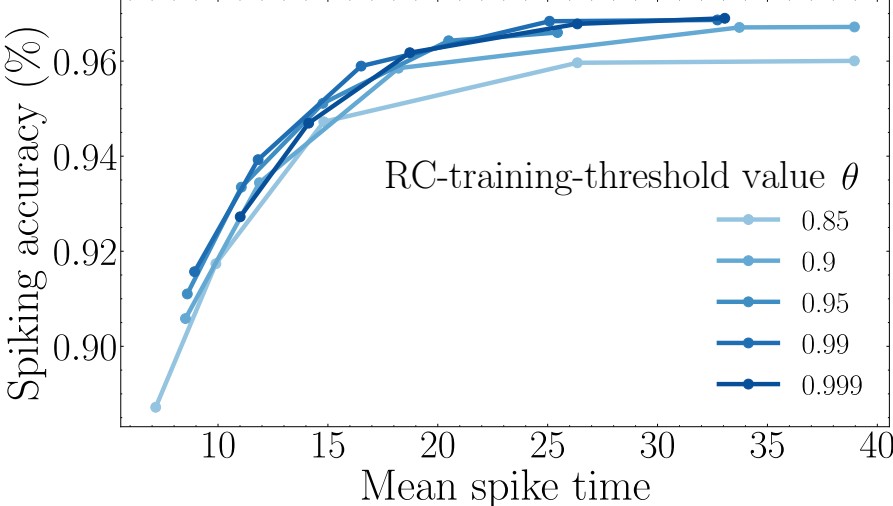

Figure 7: Effect of the threshold used during the RC training-phase on the 2-sequence problem's test performance. Colours and dots on a curve represent training-phase- and testing-phase-$\theta$ respectively.

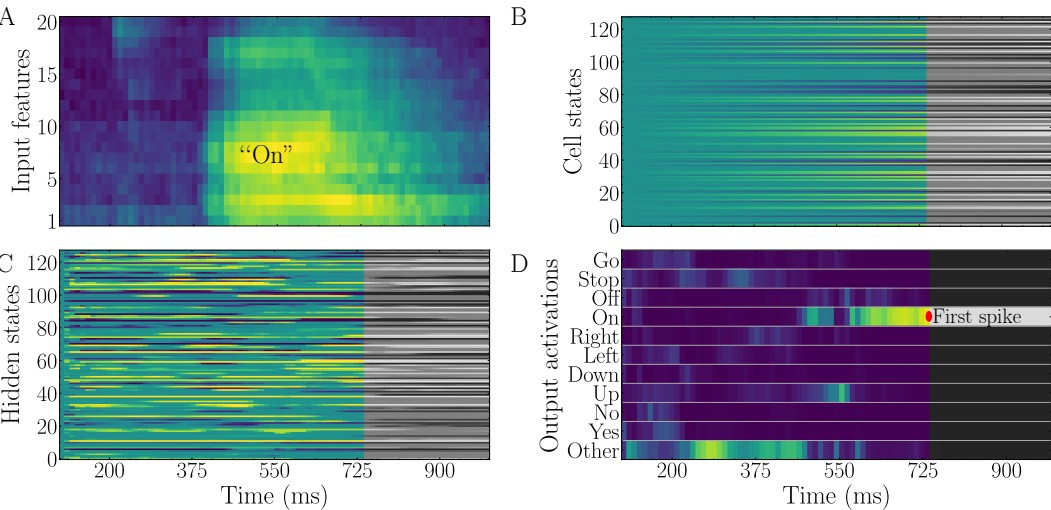

Figure 8: Hidden temporal dynamics in the RC LSTM network in response to an example recording of the utterance "On". The activity that would occur if operation were not stopped after the first output spike is shown in grey-scale.

## E   RC TRAINING IN FULLY-SPIKING NETWORKS (SNNS)

Our aim in this paper was to explore the possibility of using the spike-inspired concept of rank coding in non-spiking networks, and reaping benefits from both worlds. We hope that our spike-inspired method could feed back to the field of SNNs in an adaptation of our scheme for fully spiking networks. In this section we study the feasibility of this in preliminary experiments. A simple and commonly used example of a spiking neuron is the Leaky Integrate and Fire (LIF) model. In discrete time, the LIF can be seen as a recurrent ANN unit which can also be trained using an adaptation of BPTT for the non-differentiable spiking activation functions. Here we implement a network of LIF neurons trained with this adaptation of BPTT, and we combine it with our RC training on the temporal MNIST task (Section 3.3). The specific implementation of LIF and adaptation of BPTT that we chose to use with RC is the one from Wu et al. (2018) who also provide a code implementation. It should be noted that in their work, the authors did not attempt to make the model learn to perform fast inference, and they did not test it on temporal MNIST, but rather on the

standard, static MNIST that was input as a constant rate code over a time window. The network was then evaluated and trained over the whole length of each input and consequent output spike sequence.

Here instead we apply our RC scheme by using only the first output spike to indicate the time when the standard cross entropy loss should be calculated from the output layer's membrane potentials and the label. We then applied BPTT adapted to the LIF exactly as described by the authors. As in Section 3.3, we trained both an RC model and a non-RC baseline which calculated the loss on the final time-step.

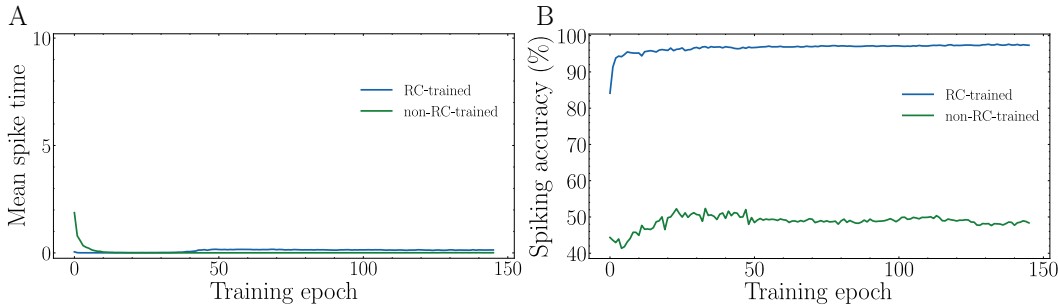

Figure 9: RC vs non-RC training of the SNN model introduced by Wu et al. (2018). A. The mean time of the first observed spike during training. B. The accuracy when evaluated at those first spikes.

The architecture we used was convolutional. Specifically, we applied this to a two-layer architecture both of 32 channels and $3 \times 3$ kernels followed by a fully-connected layer of 128 neurons. Padding of size one was applied after each convolutional layer. We selected the hyper-parameters of the model by first training a spiking CNN model using the rate coding scheme described in Wu et al. (2018) and matching the paper's reported accuracy on MNIST. We then used these same choices when training our two models from scratch. We used a threshold of 0.5, a decay factor of 0.2, a derivative approximation factor of 0.5, batch size of 400 and learning rate of 0.001.

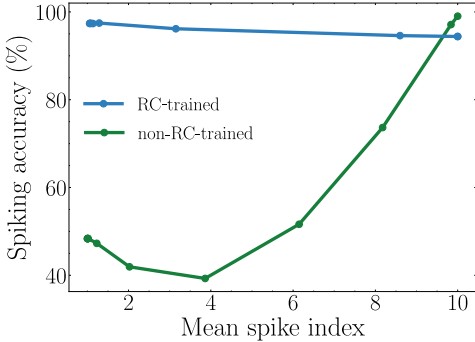

Figure 10: Speed-accuracy trade-off curves of the final SNN models when evaluated at a range of threshold values.

The results of training are included in Figure 9 with the final models evaluated at a range of thresholds in Figure 10. Since the model is designed to use a spike-coding scheme, it is not surprising to observe that both models quickly learn to fire the first spike at the beginning of the sequence (Figure 9 A). Interestingly, however, the same behaviour is observed as in the ANN case where the RC-trained model provides much more accurate classifications in those early spikes (Figure 9 B). We also consider varying the threshold on the final trained model where we observe that the low RC-accuracy of the non-RC-trained model can be improved by increasing the threshold value and encouraging a slower inference speed, as expected. On the other hand, in contrast to the ANN models, the RC-trained model here seems to decrease accuracy slightly when the threshold is increased which may highlight some fundamental differences between these two neural network models.

All in all, our RC training appears applicable to fully spiking models too.

