# OpenReview forum: "Spike-inspired rank coding for fast and accurate recurrent neural networks"
_ICLR.cc/2022/Conference — ICLR 2022 Spotlight_

### Official Review · Reviewer_c3tt · 2021-10-29

**Correctness:** 3
**Technical Novelty And Significance:** 3
**Empirical Novelty And Significance:** 3
**Recommendation:** 8
**Confidence:** 4

**Main Review:**

The paper is very clearly written and has an extensive introduction that allows to introduce the scientific problem. The method is briefly described and then applied in experimental demonstrations. If the first example is particularly simple, the three following applications are more challenging and show the advantage of using this method compared to the state of the art. One. A particularly interesting point is the trade-off curves between speed and accuracy as well as the performance of the network depending on the use or not of rank coding.

I would like to point out some limitations of this work and how it could be improved. First of all, it seems that the rank coding used in this paper is radically different from the one proposed by Thorpe and Gautrais. Indeed, in the latter an analog vector is transformed into a vector in which its analog values are ordered from the highest to the lowest. This representation has many properties such as being invariant to continuous monotonic transformations of the analog values, as for example an image can be transformed by a change of its contrast. This transformation also keeps a very high complexity of possible representations which corresponds to the set of permutations of all ranks, that is to say, to the factorial of the dimension of this vector. However in this paper it seems that you only consider the maximum of the analog vector when it exceeds a threshold. This indeed allows to transform the calculation into a temporal calculation, but the complexity of this operation is much lower than the original rank coding.


**Summary Of The Paper:**

This paper presents the original idea of applying rank coding to classical LSTM networks in order to improve their performance.

**Summary Of The Review:**

In the first practical example it seems that you use this coding in different time windows that correspond to different bins in a sequence and that these sequences are then processed independently. This information processing is far too simplified to qualify as temporal coding and thus to correspond to a spiking neuron model. The other experiments seem to show a clear advantage to the rank coding and seem promising to extend it to the coding of the maximum but also of the successive values. I encourage the authors to apply this kind of method to larger images as it has already been done in the literature.

---

> ### Author Response · Authors · 2021-11-13
> **Response to Reviewer c3tt**
>
> **1. it seems that the rank coding used in this paper is radically different from the one proposed by Thorpe and Gautrais**
>
> Indeed our rank coding is different from the spiking networks of Thorpe and Gautrais. A key aspiration of this publication is indeed to stimulate the introduction to ANNs of other spike-inspired coding schemes, of which there are multiple possibilities, each with different possible benefits. Nevertheless, we view the present work as a significant first step towards merging SNN coding schemes and their benefits into ANNs.
>
> We have added a clarification to the revised manuscript’s discussion that our rank coding is different to that of Thorpe and Gautrais.
>
>
> **2. you only consider the maximum of the analog vector when it exceeds a threshold. This indeed allows to transform the calculation into a temporal calculation, but the complexity of this operation is much lower than the original rank coding**
>
> Our implementation is indeed much less complex by applying a simple rule based on a threshold. This can be seen as an advantage, but we understand also the drawbacks compared to a more complex implementation. However, the network manages to conform to this simple rule through more complex operations that do implement a complex temporal code in the recurrently connected population of neurons. Essentially, the simplicity is in the supervisory interface for training the network, but, to map the input sequence to a well-timed and accurate summarizing spike, the network uses a more involved, multidimensional, i.e. multi-neuron, temporal encoding in the neurons’ hidden state and cell state.
>
>
> **3. In the first practical example it seems that you use this coding in different time windows that correspond to different bins in a sequence and that these sequences are then processed independently. This information processing is far too simplified to qualify as temporal coding and thus to correspond to a spiking neuron model.**
>
> We agree that the interesting aspect in this example is not at the coding of the input, which is generated as a single-channel, discrete-time Poisson spike train of a fixed rate. However, it is a proof of concept of the timing and sparsity advantages of our RC mechanism, and it provides insight into how the RC-trained network’s computations differ from the standard ones (Figure 1 of the manuscript).
>
> **4. I encourage the authors to apply this kind of method to larger images as it has already been done in the literature.**
>
> Here we used the (temporal) MNIST dataset to be able to compare with the relevant literature, as it is the main benchmark that is used in SOTA SNNs (see for example Google’s Comsa et al. (2021), and Zhang et al. (2021)). Of course, we did recognize the simplicity of MNIST images, and to target both a larger problem and a different domain, we applied our method to speech recognition (section 3.4).
>
> **5. The other experiments seem to show a clear advantage to the rank coding**
>
> We appreciate your ideas for future experiments, as well as your recognition of the advantages of the paper. We believe that after our clarifications here and the revisions to the manuscript, our work is sufficiently valuable for presentation to ICLR.
>
> &nbsp;
>
> **References**
>
> * Iulia-Maria Comsa, Krzysztof Potempa, Luca Versari, Thomas Fischbacher, Andrea Gesmundo, and Jyrki Alakuijala. (2021). Temporal coding in spiking neural networks with alpha synaptic function: Learning with backpropagation. IEEE Transactions on Neural Networks and Learning Systems.
> * Malu Zhang, Jiadong Wang, Jibin Wu, Ammar Belatreche, Burin Amornpaisannon, Zhixuan Zhang, Venkata Pavan Kumar Miriyala, Hong Qu, Yansong Chua, Trevor E Carlson, et al. (2021). Rectified linear postsynaptic potential function for backpropagation in deep spiking neural networks. IEEE Transactions on Neural Networks and Learning Systems.

---

> > ### Author Response · Authors · 2021-11-17
> > **Hidden temporal dynamics: manuscript revision and newly added data**
> >
> > We have updated the manuscript to include an Appendix C that provides the information we give in point 2 above, along with a supporting ~~Figure 7~~ _edit:_ Figure 8 that adds new data to the manuscript.

---

> > > ### Comment · Reviewer_c3tt · 2021-11-22
> > > **thanks for the response**
> > >
> > > Many thanks to the authors for their efforts in responding to my comments and to that of the other reviewers. Supporting Figure 8 suggests a nice development of this work.
> > > I have updated my evaluation accordingly/

---

### Official Review · Reviewer_zTtT · 2021-11-02

**Correctness:** 3
**Technical Novelty And Significance:** 3
**Empirical Novelty And Significance:** 3
**Recommendation:** 8
**Confidence:** 4

**Main Review:**

Strength:
* As far as I know the idea is new
* Well written

Weaknesses
* Experimental validations are well below modern ML standards
* The authors should test their idea on spiking neurons (eg LIF) as opposed to LSTMs

I think experimental validation falls short for ICLR. Only the Google Speech Command dataset is not toy, and the accuracy they get on this dataset is below the SOTA.

In addition, the authors seem to target the spiking neural network community (part of which uses ROC). See for example Table 1. But then, for a fair comparison, they should try to use a spiking neuron model (eg leaky integrate and fire, LIF) instead of the LSTM. In discrete-time, the LIF can be seen as a recurrent ANN unit, and BPTT could be used as well.

The authors seem to use batch processing, but I don't understand how it's possible. The number of timesteps used by BPTT is example-dependent. How can batch processing work with a varying number of timesteps? More insight is needed here.



**Summary Of The Paper:**

The authors introduce a new way to train RNNs using rank order coding (ROC). With ROC the label is given by the first readout unit to reach a threshold. As soon as this happens, the processing is stopped, and BPTT is used from that particular time step, using the predictions at that particular time step and the ground truth. This will encourage the neuron with the right label to be as active as possible at that particular time step, and thus its threshold will tend to be reached earlier in the future. This is desirable, as the latency of the decision will decrease. Furthermore, the speed-accuracy trade-off is tunable by varying the threshold.

The authors validate their idea using LSTMs on two toy problems, and then on MNIST and on the Google Speech Command dataset.

**Summary Of The Review:**

A potentially interesting idea, but not yet validated

---

> ### Author Response · Authors · 2021-11-13
> **Response to Reviewer zTtT (part 1)**
>
> **1. Experimental validations are well below modern ML standards […] Only the Google Speech Command dataset is not toy and the accuracy they get on this dataset is below the SOTA**
>
> Whilst we appreciate the review offered to our work, we strongly disagree that our experimental validations fall well below modern ML standards or fail to validate our idea.
>
> _Regarding SOTA performance:_
>
> Firstly, we would like to note that our paper is indeed advancing the SOTA in the temporal coding literature. We outperform previous temporal coding networks in the principal benchmarking task of the literature, namely temporal MNIST (see e.g. Google’s Comsa et al. (2021)). Taking the temporal coding field’s SOTA further, we apply it to real-world speech data and show benefits compared to conventional RNNs. We believe that these make our work valuable to the large community of neuromorphic computing.
> It should be noted that biologically-inspired approaches often fail to achieve accuracies of conventional ones. Therefore our aim was not to improve SOTA accuracy, but to introduce a biologically-inspired mechanism for speed and efficiency. Despite this, interestingly our top accuracy does achieve and slightly surpass the literature’s SOTA accuracy among RNNs on Google Speech Commands and, in addition to that, it offers important improvements in speed, efficiency, and tunability of tradeoffs.
>
> Specifically, we achieve 95.6% correct word recognition while only using a portion of the full sequences (~65% on average at inference time). On the other hand, the RNN SOTA for this task is reported in the influential paper “Hello Edge” by Amazon’s Zhang et al. (2017), who achieved a top accuracy of 94.8% with an LSTM, 95% with a CRNN, and 95.4% with even a convolutional network DS-CNN and do not attempt to achieve our speedups. Our slight advantage in accuracy may be explained by faster inference relying on shorter-term dependencies but this is not a central benefit we wish to present.
>
> Some more recent non-recurrent approaches (e.g. Kim et al. (2021)) do achieve higher accuracies on this task, as attention-based and residual neural networks have emerged as quite successful in this domain, which has been the case in other domains as well. RNNs, however, may have certain inherent strengths for streaming data that justify continued research on them. As a matter of fact, non-RNN approaches applied along a sliding window rarely report performance on the actually intended application of live streaming audio, but in the very recent case where this was tested and reported (i.e. in the resnet of López-Espejo et al., 2021), there is a significant drop in performance (>10%) between audio clips and streaming input. Therefore, improvement of RNNs remains a key goal and RNNs are still under heavy research and use, including in the specific application of keyword spotting. For example, very recently, on the dataset of Google Speech Commands, and, as in our case, with the aim of reducing delays, Tan et al. (AAAI 2021, cited in our manuscript) also used an LSTM in combination with their technique. In summary, we believe that our results on Google Speech Commands are very much relevant to the current SOTA, and offer additional important improvements in terms of inference speed and efficiency.
>
> _Regarding our range of experimental demonstrations:_
>
> We disagree that our demonstrations are well below modern standards for modern ML in general or even the high standards of ICLR in particular. For example, we highlight a number of high-quality papers, accepted in last year’s ICLR, that present a similar or smaller scale of experimentation as ours under comparable circumstances, and more exist.
>
> * Menick et al. (2021) from DeepMind and Google, in an ICLR 2021 spotlight paper present a practical method for learning parameters of recurrent neural networks in an online fashion. Here the authors’ experimental validations consist of a synthetic task (copy task) and a single real world language-modelling task (WikiText103).
> * Stanford’s Sahiner et al. (2021) propose a convex dual network for the task of image reconstruction. Their method is evaluated on MNIST and a fastMRI dataset.
> * Similarly, Zhou et al. (2021) validated their method on two datasets, which were sequences of events, either synthetically generated or recorded from neuronal populations.
>
> Our work presented four experiments, and each experiment was carefully selected. For example, our first two demonstrations use sufficiently controlled data to offer different insights into the workings of our mechanism, but still in problems difficult enough for a conventionally trained LSTM to solve as well as our mechanism does.
>
> _continued in next comment..._

---

> > ### Author Response · Authors · 2021-11-13
> > **Response to Reviewer zTtT (part 2)**
> >
> > _...continued from previous comment_
> >
> > We then used the temporal MNIST task to be able to compare with the relevant literature, as it is the main benchmark that is used in SOTA SNNs (see for example Google’s Comsa et al. (2021), and Zhang et al. (2021)). It should be noted that, despite their controlled setting, these three tasks are not easy, as they have to combine long-term memory, on-line adaptability, and speed. Therefore, we suggest that four such tasks, including a real-world application such as Google Speech Commands, is not a limited set of demonstrations for a single paper. Rather it achieves a difficult balance between mechanistic insight, interdisciplinary comparison with SNNs, and real-world applicability. Importantly, it is of the high standards that other recently accepted ICLR papers set.
> >
> > We thank the reviewer for raising this query and based on this we have revised the manuscript by adding to the Appendix further explanation of our motivations for each experiment to clarify what it intends to evaluate. Overall, we believe that the insights we provide, the range we cover, and the advantages we show are sufficient for publication and to be adopted by the community.
> >
> >
> >
> > **2. The authors should test their idea on spiking neurons (eg LIF) as opposed to LSTMs. […] they should try to use a spiking neuron model (eg leaky integrate and fire, LIF) instead of the LSTM**
> >
> > Our aim in this paper was to test the possibility of using this spike-inspired concept in non-spiking networks, and reaping benefits from both worlds. We hope to establish this as the goal in the following passage from our paper’s introduction:
> >
> > > Taking together the limitations and benefits of SNNs and of other approaches, what is needed is a strategy that introduces aspects of RC into conventional, non-spiking ANNs, including during training. This would potentially reap the speed and efficiency benefits of this neuromorphic scheme, without abandoning the real-world usability and performance of ANNs. In addition, these insights into neural coding could feed back to neuroscience. Here we describe and demonstrate such a strategy for temporal coding in deep learning and inference with ANNs. In addition, these insights into neural coding could feed back to neuroscience.
> >
> > We believe that this stated objective is indeed valuable to the related communities, and using LIF neurons instead of LSTMs would not have gone as far towards achieving this goal as we did. However, we certainly agree that using our insights from ANNs to feed back again to more biologically-realistic neurons such as LIF is one of the next steps that are motivated by this paper.
> >
> > In our opinion, the results using LSTMs make our work more broadly valuable than only to the SNN community, so we suggest that the paper is worthy of publication with its current content.
> >
> > **3. How can batch processing work with a varying number of timesteps?**
> >
> > In batch training mode, apart from the resulting inference speedups, there are three training efficiencies from RC, of which some are indeed not immediately obvious how to achieve. First, in the most naïve implementation, the full batch and the full sequences are passed forward through the network. However, backpropagation at each batch element starts from only an intermediate time-step that corresponds to the earliest spike. Therefore the backward computations are truncated at an intermediate layer of the deep unrolled RNN for each element in the batch, speeding up the computation. In a slightly more efficient implementation and in addition to the latter benefit, the outputs of the RNN are observed on-line, at every time step of the batch’s sequential processing, and as soon as all batch elements have resulted in at least one spike, the forward pass is stopped. Finally, the maximally efficient implementation of RC training can be achieved by masking the batch elements at each time-step if they have already previously resulted in a spike. Then the effective batch size becomes progressively smaller throughout the sequence, thus not only ending the batch sooner, but also speeding up the processing of each time step. That implementation is similar to this concept: https://stackoverflow.com/a/56211056, but in lower-level code customization (e.g. cuda), to account for the fact that the sequence lengths are not known in advance.

---

> > > ### Author Response · Authors · 2021-11-13
> > > **Response to Reviewer zTtT (references)**
> > >
> > > **References**
> > > * Kim, B., Chang, S., Lee, J., & Sung, D. (2021). Broadcasted Residual Learning for Efficient Keyword Spotting. arXiv preprint arXiv:2106.04140.
> > > * López-Espejo, I., Tan, Z. H., & Jensen, J. (2021). A Novel Loss Function and Training Strategy for Noise-Robust Keyword Spotting. IEEE/ACM Transactions on Audio, Speech, and Language Processing, 29, 2254-2266.
> > > * Menick, J., Elsen, E., Evci, U., Osindero, S., Simonyan, K., & Graves, A. (2020). Practical Real Time Recurrent Learning with a Sparse Approximation. In International Conference on Learning Representations.
> > > * Sahiner, A., Mardani, M., Ozturkler, B., Pilanci, M., & Pauly, J. (2020). Convex regularization behind neural reconstruction. In International Conference on Learning Representations.
> > > * Zhou, F., Zhang, Y., & Zhu, J. (2020). Efficient Inference of Flexible Interaction in Spiking-neuron Networks. In International Conference on Learning Representations.
> > > * Iulia-Maria Comsa, Krzysztof Potempa, Luca Versari, Thomas Fischbacher, Andrea Gesmundo, and Jyrki Alakuijala. Temporal coding in spiking neural networks with alpha synaptic function: Learning with backpropagation. IEEE Transactions on Neural Networks and Learning Systems, 2021.
> > > * Xinrui Tan, Hongjia Li, Liming Wang, Xueqing Huang, and Zhen Xu. Empowering adaptive early-exit inference with latency awareness. In Proceedings of the AAAI Conference on Artificial Intelligence, volume 35, pp. 9825–9833, 2021.

---

> > > > ### Comment · Reviewer_zTtT · 2021-11-15
> > > > **Still not convinced**
> > > >
> > > > I appreciate the authors' response. But I'm still not convinced. In my opinion, in its current state the manuscript is not convincing enough:
> > > >
> > > > * I don't think the deep learning community will be convinced by an approach whose accuracy (95.6%) is significantly below the SOTA (98.6%) on the only non-toy dataset, Google Speech Command (https://paperswithcode.com/sota/keyword-spotting-on-google-speech-commands). I acknowledge that they do not process all the timesteps, but I don't think this advantage is worth the drop in accuracy.
> > > >
> > > > * I don't think the SNN community will be convinced, because, as I said in my first review, the authors used LSTMs and not spiking neurons.
> > > >
> > > > Therefore I maintain my rating: "5: marginally below the acceptance threshold"

---

> > > > > ### Author Response · Authors · 2021-11-15
> > > > > **2nd response to Reviewer zTtT**
> > > > >
> > > > > We appreciate the response, but we believe it does not account for the points made in our previous comment, which we summarise again here.
> > > > > 1) Regarding the deep learning community:
> > > > > - RNNs are worth exploring more even if alternative approaches outperform them.
> > > > > - SOTA accuracy was not the goal - speed and efficiency for RNNs was. Nevertheless, we achieved RNN SOTA accuracy.
> > > > > - We provide speedups that the literature shows are clearly important to many researchers, in the field of keyword spotting and more generally to RNNs and deep learning, even if they don't interest the reviewer.
> > > > >
> > > > > _Rejecting this paper on this basis would imply rejecting any future progress in RNNs or low-footprint computation, because attention- or resnet-based models are more accurate under some unconstrained conditions._
> > > > > ***
> > > > > ***
> > > > >
> > > > > 2) Our work with LSTMs is certainly interesting to the SNN community:
> > > > > - We have provided SOTA temporal coding results on that community’s benchmark task (and circumvented several of SNNs’ difficulties).
> > > > > - We have extended the benefits and usability of that community’s temporal coding concepts into any conventional RNN/LSTM.
> > > > >
> > > > > _We believe that merging SNN concepts with conventional models like LSTMs should be encouraged, or at least not hindered by such demands, for the benefit of both communities._

---

> > > > > > ### Comment · Reviewer_zTtT · 2021-11-22
> > > > > > **Interesting for the RNN community.**
> > > > > >
> > > > > > Fair enough, the authors convinced me that this work is interesting for the RNN community. So I've raised my score to 6.

---

> > > > > > > ### Author Response · Authors · 2021-11-23
> > > > > > > **SNN experiments added**
> > > > > > >
> > > > > > > Thank you for the constructive discussion.
> > > > > > >
> > > > > > > We have now further updated the manuscript with preliminary SNN experiments as well in Appendix D.
> > > > > > >
> > > > > > > We apply RC-training on a LIF model, and we observe similar advantages from RC as in the ANNs, once the common difficulties associated with simulating and training spiking models are surpassed.

---

> > > > > > > > ### Comment · Reviewer_zTtT · 2021-11-24
> > > > > > > > **POC with SNN is much appreciated**
> > > > > > > >
> > > > > > > > The proof of concept with SNNs, which have been the main use-case of rank-order coding so far, is much appreciated.
> > > > > > > > I've raised my score to 8.
> > > > > > > >
> > > > > > > > I'm curious: in the hidden layers, activity is not limited to one spike per neuron, is it?

---

> > > > > > > > > ### Author Response · Authors · 2021-11-24
> > > > > > > > > **Thank you for leading to SNN-related added value**
> > > > > > > > >
> > > > > > > > > We agree that this added POC further increases the paper's value for the SNN community, so thank you for the suggestion.
> > > > > > > > >
> > > > > > > > > To your question: We did not explicitly limit the hidden layer's spikes, or otherwise change the network's operation, so hidden neurons can indeed spike more than once before the first spike of the output layer.

---

### Official Review · Reviewer_dNmj · 2021-11-06

**Correctness:** 4
**Technical Novelty And Significance:** 3
**Empirical Novelty And Significance:** Not applicable
**Recommendation:** 8
**Confidence:** 3

**Main Review:**

PROS:

I think the proposed method is very practical and the ideas of this paper are organized logically.

The method is related to early-exit inference but their model not just exits early, it does so by a learned rank code (RC) that is inspired by spiking neural networks. This allows for adaptively decreasing computation and increasing speed during both training AND inference.

For training, the idea of backpropagating from a strategically early time step of each sequence, determined by the time at which an output neurons activation crosses a threshold, is interesting and can lead to significantly shorter training time and lower compute resources.

It appears that prior and related work is adequately referenced throughout the paper. Unfortunately I am not familiar enough with this line of work, but I quick search revealed no glaring omission.

The empirical methodology appears standard, and is reported in sufficient detail to recreate the results. The resulting claims are justified by the performance of the method. Strengths and limitations are sufficiently discussed. The writing is clear and succinct.

CONS:

The authors do not compare their method to non-RC trained LSTMs in the temporal MNIST task. For this task, since it appears that the first frame contains most of the information, I suspect that an early-exit but non-RC trained LSTM would perform similar in terms of inference speed and accuracy. This is however just a guess, did the authors try this or have further insights? Comparing their model to SNNs in this task seems somehow unfair, also given that the authors state earlier that “ANNs are simpler to train, and usually achieve superior performance”.
However, I can follow the authors argumentation on why they have chosen to compare to SNNs. Comparing to early-exit but non-RC trained LSTMs would still be interesting but omitting it would not weaken the message delivered in the overall strong paper.

The threshold is a fixed hyper-parameter of the model. Would it be beneficial to learn this parameter?


**Summary Of The Paper:**

The authors propose a method for fast and efficient classification of sequential data. The guiding principle is that for some data modalities it is not necessary to see the whole sequence in order to make a fairly certain classification. Their model reduces inference time by learning a rank code that is inspired by spiking neural networks. Reported results show improved inference times in two toy sequence classification tasks, temporal MNIST, and in Google Speech Commands classification (compared to models without optimizing timing of inference through learning a rank code). Increasing inference speed comes with a minimal decrease in accuracy, the authors, however, introduce and show the effectiveness of a regularization term that allows for tuning of this speed-accuracy trade-off.

**Summary Of The Review:**

Overall, I vote for accepting. I like the idea of integrating specific isolated aspects of biological neurons into otherwise conventional ANNs. Here, the authors show that ANNs can benefit by such an approach and I think that further exploration of such methods may advance both spiking neural network and conventional deep learning research.

---

> ### Author Response · Authors · 2021-11-13
> **Response to Reviewer dNmj**
>
> **1. I suspect that an early-exit but non-RC trained LSTM would perform similar in terms of inference speed and accuracy. This is however just a guess, did the authors try this or have further insights?**
>
> We did not try this in this task but, based on the other experiments, which also include sufficient information early in the sequence, we do not expect a non-RC-trained LSTM to perform inference as fast as the RC-trained one, even if RC-inference is used. That is because the non-RC training objective is to maximize accuracy specifically at the end of each input sequence. Moreover, as can be seen in the literature we compare to, even temporally coded SNNs do not necessarily learn the fast regime. However, it is true that the temporal MNIST dataset has the particularity that you astutely observe, so it would be interesting to see if the LSTM can exploit it to some extent without RC training. If we complete this experiment during this discussion period, we will report the result here and also add it to the paper.
>
> _edit: We have now performed the experiment. See [separate comment](https://openreview.net/forum?id=iMH1e5k7n3L&noteId=aBJ-z9ALJF) and updated Appendix A.3._
>
>
> **2. Comparing their model to SNNs in this task seems somehow unfair, also given that the authors state earlier that “ANNs are simpler to train, and usually achieve superior performance”. However, I can follow the authors argumentation on why they have chosen to compare to SNNs.**
>
> Indeed, we believe it is important to include a comparison to SNNs for the completeness of the paper. Temporal MNIST is the benchmark task for temporal coding approaches and these have previously been contained within the SNN literature (see e.g. Zhang et al. (2021)). While comparisons between SNNs and large scale ANNs would indeed be an unfair comparison, we note that many of the existing temporal coding works from the SNN literature do themselves compare to ANN networks with comparable architectures and the overall intention of the field is to outperform ANNs by certain metrics, e.g. speed. For example, Google’s Comsa et al. (2021), and Kheradpisheh et al. (2019) both compare their proposed approaches to traditional ANNs on the temporal MNIST task with the former concluding that “generally-speaking, the performances of the two types of networks are comparable”, referring to the accuracy, but proposing the SNN as a path to faster inference. Therefore, our comparison to SNNs is a necessary part of that ongoing discussion in the literature.
>
>
> **3. The threshold is a fixed hyper-parameter of the model. Would it be beneficial to learn this parameter?**
>
> This is something we considered. In order to give a detailed response, we would like to run some additional experiments and we will reply with those results upon their completion.
>
> &nbsp;
>
>
> **References**
> * Iulia-Maria Comsa, Krzysztof Potempa, Luca Versari, Thomas Fischbacher, Andrea Gesmundo, and Jyrki Alakuijala. (2021). Temporal coding in spiking neural networks with alpha synaptic function: Learning with backpropagation. IEEE Transactions on Neural Networks and Learning Systems.
> * Malu Zhang, Jiadong Wang, Jibin Wu, Ammar Belatreche, Burin Amornpaisannon, Zhixuan Zhang, Venkata Pavan Kumar Miriyala, Hong Qu, Yansong Chua, Trevor E Carlson, et al. (2021). Rectified linear postsynaptic potential function for backpropagation in deep spiking neural networks. IEEE Transactions on Neural Networks and Learning Systems.
> * Kheradpisheh, S. R., & Masquelier, T. (2020). Temporal backpropagation for spiking neural networks with one spike per neuron. International Journal of Neural Systems, 30(06), 2050027.

---

> > ### Author Response · Authors · 2021-11-17
> > **Tuning the training threshold**
> >
> > Regarding your question about learning the threshold hyperparameter that is used during training, we found that tuning it has only a minimal effect on performance, as long as the threshold is reasonably high. We have added a section B to the Appendix of the revised manuscript, with a ~~Figure 6~~ _edit:_ Figure 7 that shows this in one of the tasks. Other ideas for future exploitation of the threshold parameter to improve performance are included in the last paragraph of section 3 of the main manuscript.

---

> > > ### Comment · Reviewer_dNmj · 2021-11-24
> > > **Thanks for the responses and additional experiments**
> > >
> > > Many thanks to the authors for responding to my comments and for performing the additional experiments. I originally thought that this paper should be accepted; I stand by my score.

---

> > ### Author Response · Authors · 2021-11-20
> > **Non-RC LSTM on Temporal MNIST is now also tested**
> >
> > > The authors do not compare their method to non-RC trained LSTMs in the temporal MNIST task. [...] Comparing to early-exit but non-RC trained LSTMs would still be interesting but omitting it would not weaken the message delivered in the overall strong paper.
> >
> > Thank you for the suggestion, this was indeed an interesting control. We have now performed this experiment too and included the results in the updated manuscript's Appendix A.3, and Figure 6. In agreement with the other experiments, the comparison confirms that it is the RC-training that causes the early inference, and not the specific dataset or the LSTM model alone.

---

### Author Response · Authors · 2021-11-24
**Concluding summary by the authors**

We would like to thank the reviewers for the constructive discussion. Based on it we have updated the manuscript with the requested new experiments, added clarifications and corrections of some typos. We are thankful for the process that allowed the paper to converge to the same recommendation by all reviewers.

---

### Decision · Program_Chairs · 2022-01-20

**Decision:**

Accept (Spotlight)

**Comment:**

The authors propose a rank coding scheme for recurrent neural networks (RNNs) - inspired by spiking neural networks - in order to improve inference times at the classification of sequential data. The basic idea is to train the RNN to classify the sequence early - even before the full sequence has been observed. They also introduce a regularisation term that allows for a speed-accuracy trade-off.

The method is tested on two toy-tasks as well as on temporal MNIST and Google Speech Commands.

The results are very good, typically improving inference time with very little loss in accuracy.

Furthermore, the idea seems novel and the paper is well written.

An initial criticism was that experiments with spiking neural networks (SNNs) were missing. The authors added a proof of concept for SNNs, which satisfied the reviewer.

The authors also added some control experiments in response to the initial reviews, which improved the manuscript.

In summary, the manuscript presents a valuable novel idea with good experimental verification and interesting aspects both for ANNs and SNNs. The reviewers consistently vote for acceptance.